# Effect of Retained Austenite and Non-Metallic Inclusions on the Mechanical Properties of Resistance Spot Welding Nuggets of Low-Alloy TRIP Steels

**Víctor H. Vargas Cortés [1], Gerardo Altamirano Guerrero [2],\*, Ignacio Mejía Granados [1], Víctor H. Baltazar Hernández [3] 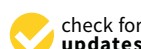 and Cuauhtémoc Maldonado Zepeda [1]**

[1] Instituto de Investigación en Metalurgia y Materiales, Universidad Michoacana de San Nicolás de Hidalgo, Edificio "U3", Ciudad Universitaria, Morelia 58030, Michoacán, Mexico; vargas-victor@live.com.mx (V.H.V.C.); imejia@umich.mx (I.M.G.); temo.maldonado@gmail.com (C.M.Z.)

[2] Instituto Tecnológico de Saltillo. Blvd, Venustiano Carranza #2400, Col. Tecnológico, Saltillo C.P. 25280, Mexico

[3] Materials Science and Engineering Program, Autonomous University of Zacatecas, López Velarde 801, Zacatecas 9800, Zacatecas, Mexico; victor.baltazar@uaz.edu.mx

\* Correspondence: galtamirano@itsaltillo.edu.mx; Tel.: +844-438-95-39; Fax: +844-438-95-16

**Abstract:** The combination of high strength and formability of transformation induced plasticity (TRIP) steels is interesting for the automotive industry. However, the poor weldability limits its industrial application. This paper shows the results of six low-alloy TRIP steels with different chemical composition which were studied in order to correlate retained austenite (RA) and non-metallic inclusions (NMI) with their resistance spot welded zones to their joints' final mechanical properties. RA volume fractions were quantified by X-ray microdiffraction (μSXRD) while the magnetic saturation technique was used to quantify NMI contents. Microstructural characterization and NMI of the base metals and spot welds were assessed using scanning electron microscopy (SEM). Weld nuggets macrostructures were identified using optical microscopy (OM). The lap-shear tensile test was used to determine the final mechanical properties of the welded joints. It was found that NMI content in the fusion zone (FZ) was higher than those in the base metal and heat affected zone (HAZ). Whereas, traces of RA were found in the HAZ of highly alloyed TRIP steels. Lap-shear tensile test results showed that mechanical properties of spot welds were affected by NMI contents, but in a major way by the decomposition of RA in the FZ and HAZ.

**Keywords:** low-alloy TRIP steel; resistance spot welding; retained austenite; non-metallic inclusions; X-ray microdiffraction and magnetometry

## 1. Introduction

Advanced high-strength steels (AHSS), particularly transformation induced plasticity (TRIP) steels are increasingly becoming used in the transportation sector due to that their high mechanical properties allowing the use of thinner gauge sheets. Consequently, the total weight of the vehicles is reduced, leading to enhanced safety and fuel efficiency. The typical TRIP multiphase microstructure is developed by the addition of alloying elements such as Al, Mn, P, and Si, in addition to a two-stage heat treatment, and it is composed of a mixture of ferrite, bainite, and metastable retained austenite. TRIP steels mechanical properties are microstructure dependent and they offer an excellent combination of strength and ductility, additionally to high energy absorption during deformation owing to the deformation-induced transformation of RA into martensite (TRIP effect) at elevated strain level [1]. Resistance spot welding (RSW) is the predominant process in sheet metal joining, particularly in the

transportation sector. Typically, there are about 2000–5000 spot welds in a vehicle [2]. Weldability of TRIP steels is affected by their rich alloying content; moreover, the designed multiphase microstructure is modified by the thermal cycle inherent to the RSW process resulting in decreased mechanical properties of the spot weld due the formation of hard phases and non-metallic inclusions. Mechanical performance in TRIP steel spot welds is affected by factors such as [3]: (i) weld nugget diameter, (ii) non-metallic inclusions development, and (iii) the decomposition of RA in the fusion zone (FZ) and heat affected zone (HAZ) after the RSW process. Non-metallic inclusions affect significantly the properties of base metals, depending on their morphology, composition, and distribution non-metallic inclusions may have detrimental effects also on the weld metal properties. Particularly, in resistance spot welded TRIP steels, the formation of hard non-metallic inclusions in the FZ and HAZ would lead to a decrease in the final mechanical properties of the spot welds.

Research works [4–7] try to attempt the development of non-metallic inclusions when welding AHSS steels, including TRIP steels, however, fail to compare the relationship between TRIP steels with different chemical compositions. In the present research work, an attempt has been made to analyze the non-metallic inclusion development when spot welding six in-lab casting and cold-rolled TRIP steels with different chemical compositions and assess their influence over the final spot weldments mechanical properties.

## 2. Materials and Methods

Six experimental TRIP steels, with different alloying contents were fabricated in the Foundry Lab of the Metallurgical Research Institute of UMSNH (Morelia, México), by using high purity raw materials, in a 25 kg capacity induction furnace. The alloying elements were added directly into the crucible, the steels were cast into 50 mm × 50 mm × 150 mm square section metal molds. The chemical composition of the TRIP steels was obtained via atomic emission spectroscopy (AES) and is shown in Table 1, together the equivalent carbon (CE) content calculated by means of Yurioka´s formula [8] and the Ac1 (temperature at which austenite formation starts during heating), Ac3 (Temperature at which austenite formation is completed during heating), and Ms (temperature at which martensite begins to form) transformation temperatures determined experimentally by quenching dilatometry. The ingots were first heat treated at 1100 °C and held for 2 h, and then air cooled. After homogenization heat treatment, the ingots were cut into rectangular sections of 25 mm × 25 mm × 150 mm and then heated up to 1000 °C, kept at this temperature for 30 min, and then hot rolled to a thickness of 4 mm, once the materials were cooled down to 300 °C, they were cold rolled to a thickness of 3 mm, as shown in Figure 1. After cold rolled, the TRIP steels were cut in longitudinal sections of 150 mm in gauge length and 30 mm in gauge width, and then mechanically rectified to a final thickness of 1.7 mm.

**Table 1.** Chemical composition (wt%), equivalent carbon and experimental transformation temperatures (°C) of the studied steels.

| Steel | C | Si | Mn | P | S | Cr | Mo | Ni | Al | Ti | V | N | $CE_Y$ | $Ac_1$ | $Ac_3$ | $M_s$ |
|-------|-----|-----|-----|-----|-----|-----|-----|-----|-----|-----|-----|-----|-----|-----|-----|-----|
| S1 | 0.266 | 0.423 | 1.407 | 0.022 | 0.0048 | 0.321 | 0.027 | 0.138 | 0.105 | 0.00040 | 0.0095 | 0.0025 | 0.503 | 726 | 823 | 366 |
| S2 | 0.274 | 0.685 | 1.428 | 0.024 | 0.0050 | 0.345 | 0.025 | 0.139 | 0.050 | 0.00072 | 0.0095 | 0.020 | 0.524 | 732 | 825 | 354 |
| S3 | 0.268 | 1.011 | 1.325 | 0.025 | 0.0063 | 0.350 | 0.026 | 0.133 | 0.082 | 0.0016 | 0.0096 | 0.016 | 0.516 | 738 | 834 | 360 |
| S4 | 0.329 | 1.388 | 1.752 | 0.021 | 0.0055 | 0.232 | 0.025 | 0.115 | 0.125 | 0.0017 | 0.011 | 0.014 | 0.625 | 739 | 826 | 330 |
| S5 | 0.215 | 1.450 | 1.373 | 0.018 | 0.0096 | 0.163 | 0.021 | 0.101 | 0.091 | 0.00094 | 0.0058 | 0.015 | 0.461 | 739 | 851 | 384 |
| S6 | 0.325 | 1.890 | 1.724 | 0.022 | 0.0084 | 0.230 | 0.028 | 0.116 | 0.283 | 0.0026 | 0.012 | 0.013 | 0.639 | 760 | 860 | 337 |

TRIP heat treatment consisted of inter-critical annealing carried out at 812 °C for 1800 s, subsequent rapid cooling at 24 °C/s in a salt bath directly from the annealing temperature to the bainite transformation temperature range and then isothermally held at 410 °C for 180 s followed by

air cooling to room temperature, as shown in Figure 1. Heat treatments were monitored through a program developed in LabView using a K-type thermocouple.

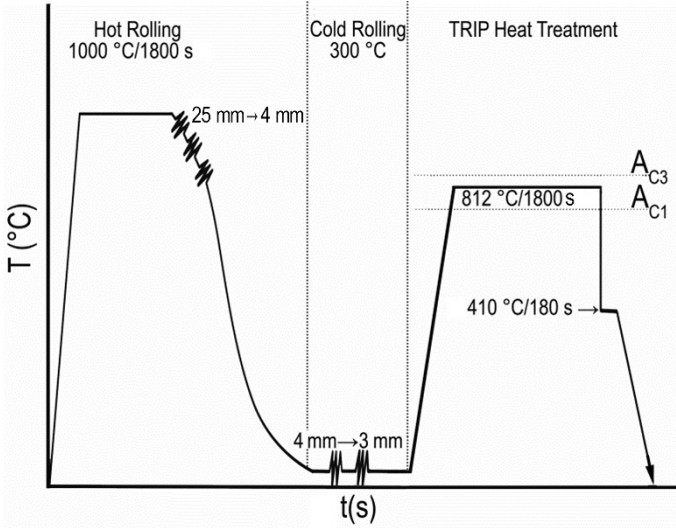

**Figure 1.** Thermomechanical transformation induced plasticity (TRIP) heat treatment of the studied steels.

Resistance spot welds were conducted in a CenterLine Ltd. 250-kVA single phase AC resistance spot welding machine, and a truncated class 2 electrode with 6.0 mm face diameter was used. The weld current was 8.3 kA, the weld force was 3.5 kN, and the weld time was 30 cycles for each material.

Samples of base metals and spot welds were metallographically prepared to a diamond-polished 0.25 μm finishing, and then chemically etched in a 2% Nital solution. Macrostructures of the TRIP steels spot welds were identified by optical microscopy (OM). TRIP steels base metals microstructures and non-metallic inclusions present in base metals and spot welds were identified by SEM. The X-ray microdiffraction (μSXRD) technique was used to quantify the microstructural phases present in the TRIP steels base metals and their welded zones. μSXRD analysis was done in sections containing the fusion zone (FZ), heat affected zone (HAZ), and base metal (BM) polished samples, using Cu-Kα radiation ($\lambda$ = 1.5402 Å). The measurements were performed in the 2θ range of 35–130°, using a step size of 0.04° 2θ, with a counting time per step of 4 s, data conditioning and analysis was processed using the TOPAS software package. The retained austenite (RA) quantitative analysis was calculated from the μSXRD intensities using the Rietveld technique and TOPAS software for automatic calculations.

In order to quantify the volume fraction of the non-metallic inclusions and retained austenite within the TRIP steels the magnetization saturation (MS) of the steel samples was calculated as a function of the applied magnetic field (H) using the magnetic saturation technique. The magnetic saturation technique measures the bulk magnetic properties of a specimen, the difference in the magnetization saturation is directly proportional to the amount of paramagnetic phases such as retained austenite + non-metallic inclusions (RA + NMI) in steels samples [9,10]. Microstructural phases such as ferrite, martensite, and cementite are ferromagnetic below their Curie temperature, while austenite and non-metallic inclusions are paramagnetic [10]. Both paramagnetic phases can be separately quantified by dissolving the RA within the sample by means of a heat treatment, in order to leave NMI as the only paramagnetic phase.

Cylindrical specimens of 1 mm in diameter were extracted from the base metals, fusion, and heat affected zones of the TRIP steels by means of an electro-discharge machine in order to not alter the RA fraction present in the steels. Magnetic measurements were performed in a Lake Shore vibrating sample magnetometer (VSM) 7307 by applying a magnetic field from one to zero and vice versa to obtain a well-defined curve, the equipment was calibrated periodically with a standard NIST (National Institute of Standards and Technology) nickel specimen. The retained austenite fraction and

non-metallic inclusions content were both calculated following the method proposed by Zhao et al. [9], in which a specimen without RA is taken as the reference one (REF) and it is compared against specimens with austenite and without austenite in order to calculate RA volume fraction and NMI contents, respectively.

Lap-shear tensile tests were carried out using a Zwick test machine with a cross head speed of 1 mm/min that is nearly quasi-static. Figure 2 shows the tensile specimen geometry and dimensions under study.

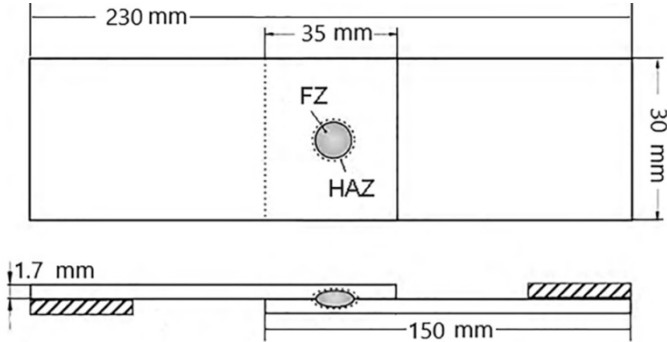

**Figure 2.** Geometry and dimensions of lap-shear tensile test specimens.

## 3. Results and Discussion

### 3.1. Microstructural Characterization of the Base Metals

Optical micrographs obtained from the base metal of TRIP steels that were under study are shown in Figure 3. Multiphase microstructures composed of ferrite (F), bainite (B), martensite (M), and retained austenite (RA) were found in all steels but in different percentages. It is evident from Figure 3 that all steels contained ferrite in the final microstructure with different portions of bainite and RA. These results confirm that the selected annealing temperature of 812 °C was within the two-phase (ferrite–austenite) inter-critical region, which was consistent with the microstructures obtained after heat treatment. For all cases, the value of the Ac3 temperature was higher than the annealing temperature used to obtain the TRIP steels. It can be seen that the RA volume fraction increased from steel S1 to S6 as the silicon content was increased too. Several investigations [11–14] have reported that in order to develop and retain austenite in TRIP steels two main factors have to be considered; the chemical composition of the steel, particularly the C, Si, Mn, Al, and P content, and in a main way by carrying out an appropriate two-stage heat treatment. Alexopoulos [15], states that through an appropriate selection of the annealing conditions, at the end of the stage the microstructure usually consists of a 50%-ferrite–50%- stable austenite mixture. The relation that rules the formation and retention of austenite between chemical composition and heat treatment parameters can be explained as follows: alloying content in TRIP steels leads to modify Ac1, Ac3, and Ms transformation temperatures, the first stage of the TRIP heat treatment consists of an inter-critical annealing between Ac1 and Ac3 to form an austenite/ferrite mixture. The inter-critical annealing temperature in the transformation zone ($\alpha + \gamma$) has been reported to be located about 20 to 30 °C above Ac1 [16] in order to develop stable RA. However, this is not generally valid for all steels. The optimal annealing temperature is a function of the chemical composition of steel and consequently of the inter-critical temperature interval. The reference point to evaluate this parameter is based on the resulting microstructure and mechanical properties. When the inter-critical annealing temperature is close to Ac3 the RA volume fraction can be high but its enrichment in carbon low. According to Yin et al. [17] the carbon concentration in austenite plays a very important role in austenitic stability, the variation between volume fraction and carbon concentration will vary for different annealing temperatures and chemical compositions. Once inter-critical annealing was done, the steels were tempered to a temperature, at which an isothermal bainitic holding was conducted, this rapid cooling caused the non-stable RA to decompose leading to

a decrease in its content. During the isothermal bainitic holding another part of the RA transformed to bainite, meanwhile the remaining austenite was stabilized due to carbon rejection from bainitic ferrite into residual austenite and to the inhibition of cementite precipitation from austenite [18].

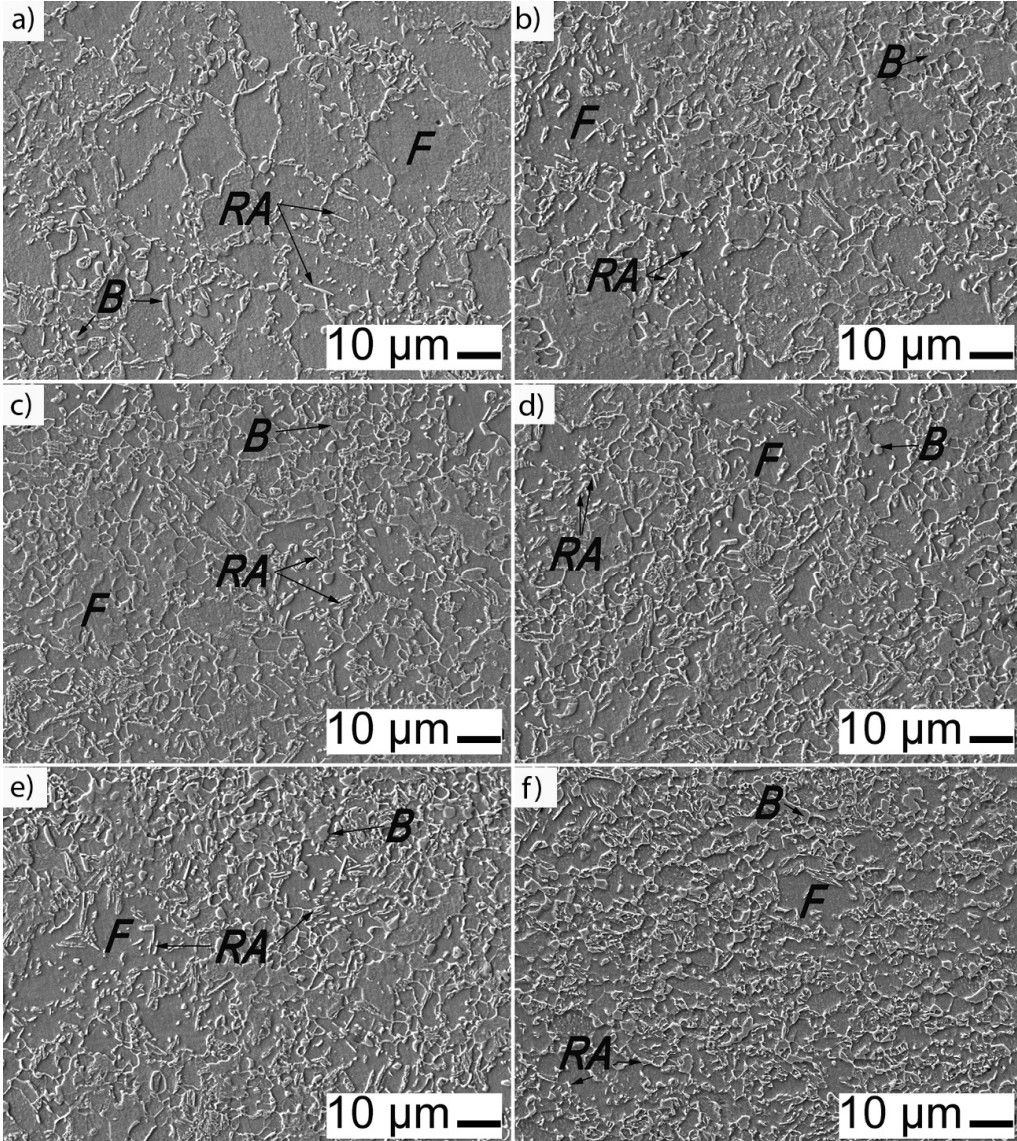

**Figure 3.** Base metal microstructure of the studied steels. (**a**): optical micrograph of S1 steel; (**b**): S2 steel; (**c**): S3 steel; (**d**): S4 steel; (**e**): S5 steel and (**f**): S6 steel.

Alloying elements such as Si, Al, and P have low solubility in cementite, so that the presence of these elements led to inhibited cementite formation, whereas the RA was enriched in carbon. It can be seen from Table 1, that the Ac3 transformation temperatures of steels S5 and S6 were high enough to promote stable retained austenite, whilst the rest of the steels (S1, S2, S3, and S4) had transformation temperatures close to that of the inter-critical annealing temperature (810 °C), so that the retained austenite developed within these steels could have been unstable.

*3.2. Microstructural Characterization of the Weld Nuggets*

Figure 4 presents the macrostructures of the spot welds. Base metals (BM), heat affected zones (HAZ) including its transformation zones as upper-critical HAZ (UCHAZ) and sub-critical HAZ (SCHAZ) and fusion zones (FZ) boundaries were clearly visible. The microstructure of the fusion

zones was composed of nearly fully martensitic phase with columnar grain growth solidification until the middle of the nugget. Heat affected zones of all spot welds seemed to have the same morphology but different sizes. Similar morphologies are obtained by Baltazar et al. [19]. The fusion zone and heat affected zones of the spot welds contained non-metallic inclusions. Apparently, the non-metallic inclusions content increased from S1 to S6 steels, in the welded zones of steels with higher amounts of Si, Al, Mn, C, and P. The distribution of the non-metallic inclusions in both zones seemed to be random in all steels. However, the density of these decreased along with increasing distance from the middle of the weld nugget. The size, morphology, and compositions of non-metallic inclusions were found to be different in all steels.

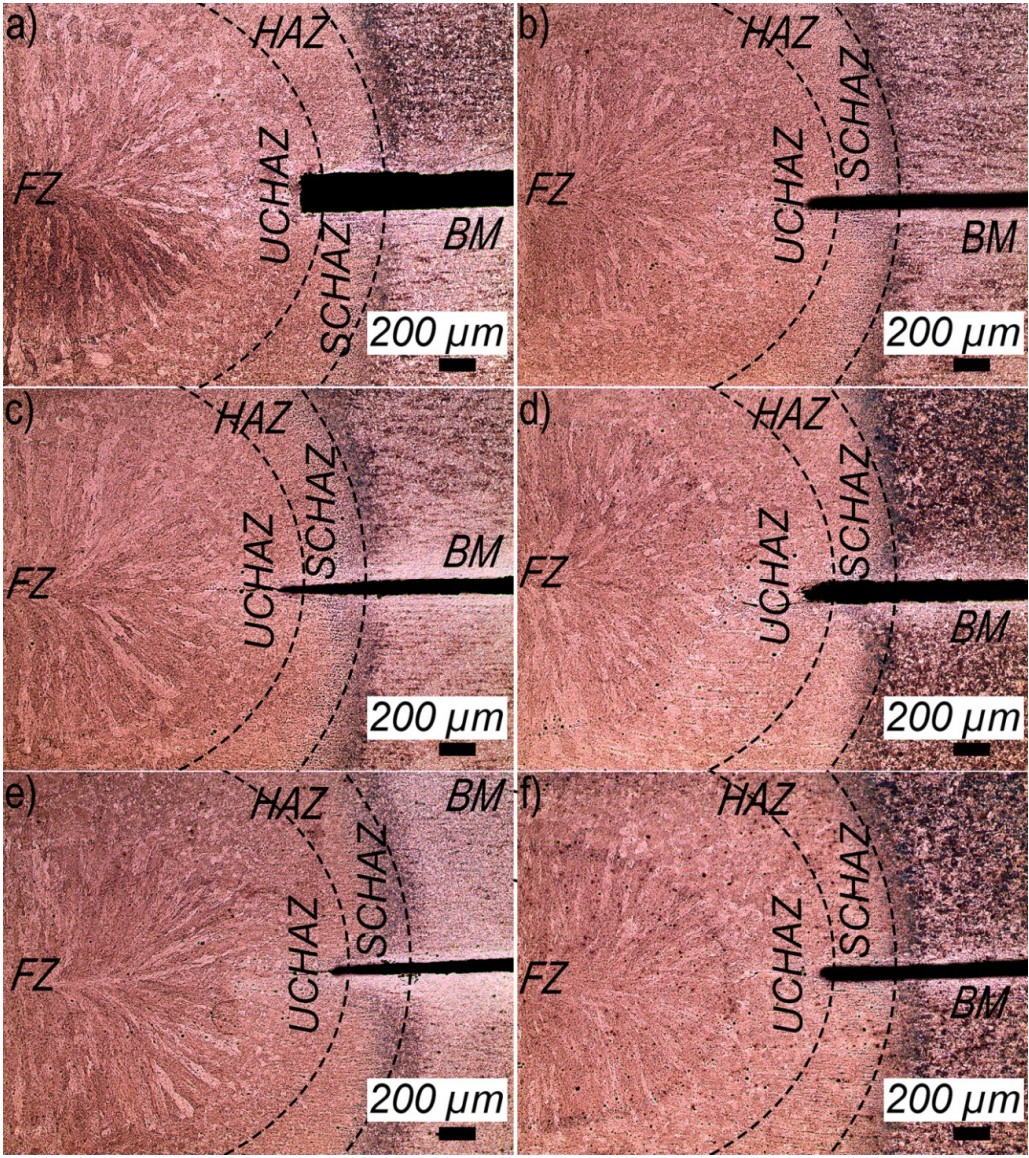

**Figure 4.** Weld nuggets macrostructures of the studied steels. (**a**): optical micrograph corresponding to S1 steel; (**b**): S2 steel; (**c**): S3 steel; (**d**): S4 steel; (**e**): S5 steel and (**f**): S6 steel.

*3.3. Quantitative Analysis of RA in the Base Metals and Spot Weld Zones*

In order to accurately quantify the microstructural phases present in TRIP steels such as RA and NMI, μSXRD and magnetic techniques samples were obtained from the base metals, fusion zones and heat affected zones of TRIP steels base metals and spot welds. By comparing both characterization techniques, the final content of NMI and RA could be properly determined in each zone. Figure 5

shows the µSXRD diffraction patterns from the TRIP steels base metals (a), heat affected zones (b) and fusion zones (c).

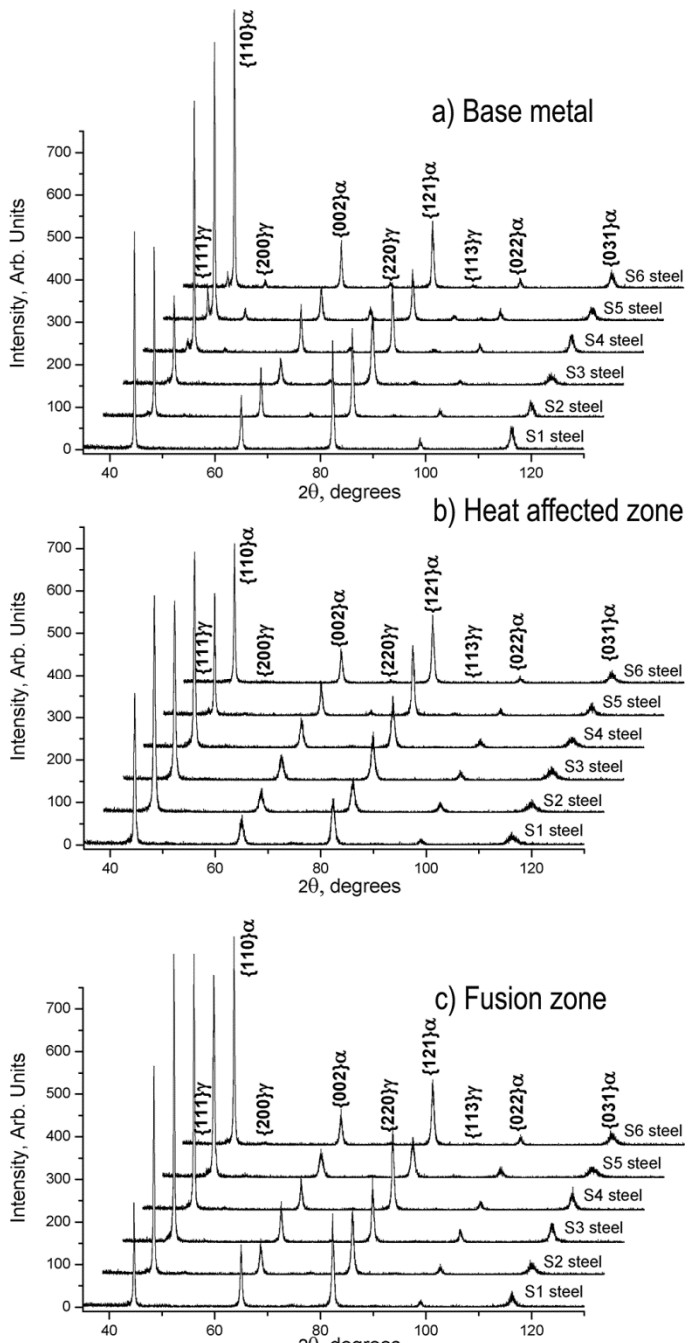

**Figure 5.** X-ray microdiffraction (µSXRD) patterns from the studied steels base metals and welded zones. (**a**): diffraction patterns obtained for base metals; (**b**): heat affected zones and (**c**): fusion zones.

Austenite (γ) and ferrite (α) peaks were clearly observed in all zones of the steels. Increments in the γ peaks intensities in the base metals were related to the retained austenite volume fraction growth, which seemed to be higher as the content of Si within the steels increased. Pichler and Stiaszny [20], reported that sufficient amounts of RA can be stabilized in TRIP steels with Si contents above 1%. However, percentages lower than 1% show less amounts of RA. After the RSW process most of RA was dissolved and only small portions can be found in the spot welded zones. As illustrated in Figure 5b, austenite (γ) peaks intensities in FZ of TRIP steels showed very little evidence of RA suggesting that

martensite was the predominate microstructure within this zone in all steels. HAZ of all steels also showed a reduction in the RA peaks intensity, this reduction seemed to be higher in the steels with lower Si content, as show in Figure 5c. S5 and S6 steels showed small RA peaks, but higher compared with the rest of the steels; this behavior pertained to a combination of RA and martensite in the HAZ. Brauser et al. [21], reported that due to the high cooling rates and based on the CCT (continuous cooling transformation) diagrams of TRIP steels only martensite can be generated within the weld button and in the HAZ adjacent to the FZ, so that the retained austenite content increases as it moves away from the FZ. This assertion differs from that reported by Nayak et al. [14], who states that there is low percentage of retained austenite in the FZ of TRIP steels welded joints.

The quantitative analysis results of the volume fraction of RA were calculated with an average error of ±0.012%, ±0.015%, ±0.015%, ±0.010%, and ±0.014% for S1, S2, S3, S4, and S5, respectively according to the TOPAS software package. Results of retained austenite volume fraction are shown in Figure 6.

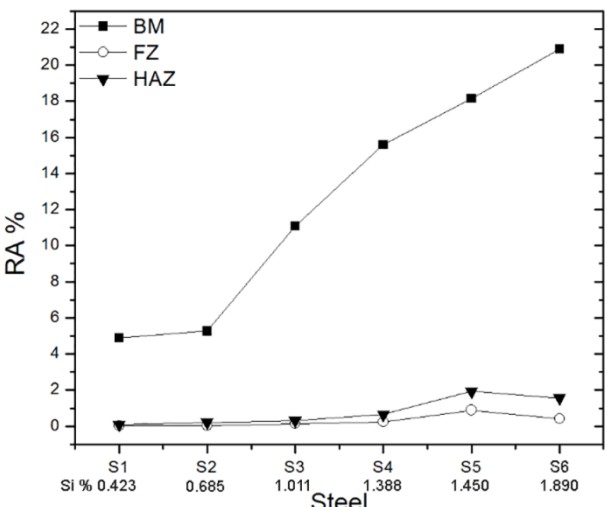

**Figure 6.** Retained austenite volume fraction in the studied steels base metals and welded zones.

As it can be seen, RA volume fractions tended to dissolve due to the high cooling rate inherent to the RSW process. However, if carbon-enriched RA was achieved by means of chemical composition and a proper TRIP heat treatment, as discussed in Section 3.1., then small amounts of RA could be present in the HAZ of the steels after the welding process, as was the case of S5 and S6 steels. These results are in accordance with that exposed by Nayak et al. [14].

### 3.4. Quantitative Analysis of NMI in Base Metals and Spot Weld Zones

NMI contents in the base metals were calculated by subtracting the RA base metals contents obtained from μSXRD from the RA + NMI volume fractions ($f_y$) base metals values, obtained by comparing the magnetization saturation of the austenite free base metals samples $M_s(f)$ against the magnetization saturation of the RA + NMI containing base metal samples $M_s(c)$ using Equation (1)

$$f_y = 1 - \beta \frac{M_S(c)}{M_S(f)}. \tag{1}$$

The volume fractions ($f_i$) of NMI within the welded zones samples of each steel were calculated by comparing the magnetization saturation of the fusion and heat affected zones samples without RA ($M_s(i)$) to their corresponding base metals samples without RA ($M_s(f)$) using Equation (2), where the coefficient $\beta = 0.99$ [9]:

$$f_i = 1 - \beta \frac{M_S(i)}{M_S(f)}. \tag{2}$$

$M_s(i)$ and $M_s(f)$ were obtained from heat treated samples at 600 °C followed by air cooling to room temperature, at this temperature the NMI present in the TRIP steels did not transform, consequently only the RA was transformed into a mixture of ferromagnetic phases, leaving the NMI as the only paramagnetic phase. The magnetization saturation method curves were obtained with relative errors of ±2%, then the linear zone of each curve was performed to a linear fitting arrangement in order to calculate $M_s(c)$ and $M_s(f)$ and their corresponding standard deviations in accordance with Reference [22] were relative errors smaller than 0.006%.

Figure 7a shows the magnetization behavior of the base metal samples without RA ($M_s(f)$) taken as references (REF) for the applied magnetic field. The magnetization behavior of the RA + NMI BM containing samples is shown in Figure 7b. It was evident that steels with higher alloying contents as Si, Mn, and C tended to decrease the mass magnetization of the steel. According with this behavior it could be seen that as the magnetization curves approached zero the RA content tended to increase. BM samples showed similar behaviors, however, representative magnetization curves were different for all steels in each condition. Table 2 summarizes the REF and BM magnetization saturation $M_s$ of all TRIP steels.

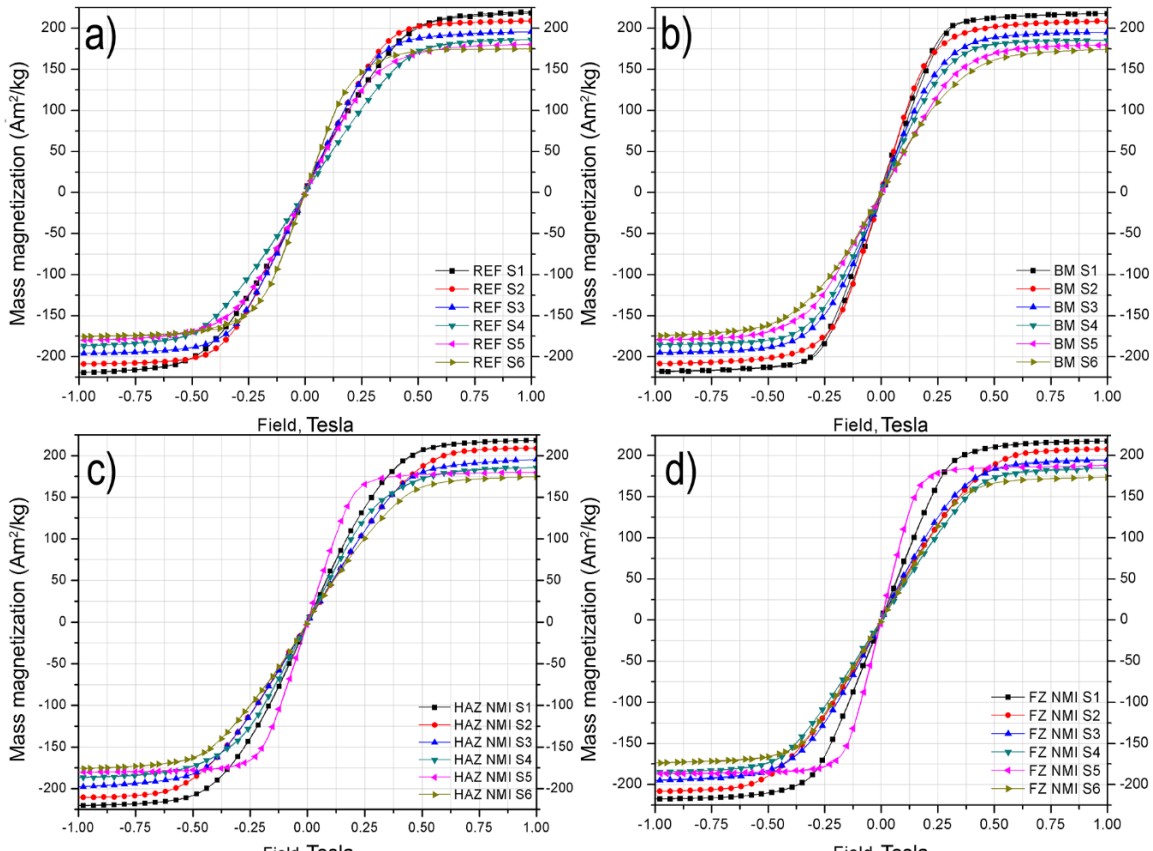

**Figure 7.** Magnetization curves of the reference (REF), base metals, and welded zones. (**a**): curves obtained for base metal samples without RA ($M_s(f)$) taken as references (REF); (**b**): curves obtained for base metal samples containing RA + NMI; (**c**): curves obtained in the HAZ of the NMI for samples steel without RA content and (**d**): curves obtained in the FZ of the NMI for samples steel without RA content.

**Table 2.** Magnetization saturation values of the austenite containing base metal (BM $M_s(c)$) and non-containing REF ($M_s(f)$) samples.

| Steel | S1 | S2 | S3 | S4 | S5 | S6 |
|---|---|---|---|---|---|---|
| BM $M_s(c)$ (Am$^2$/kg) | 206.74 | 196.9 | 172.2 | 155.9 | 146.5 | 137.1 |
| REF $M_s(f)$ (Am$^2$/kg) | 218.3 | 208.8 | 195.6 | 186.1 | 180.4 | 175.1 |

Magnetization curves of HAZ and FZ of the NMI non-containing RA samples are shown in Figure 7c,d, respectively. The magnetic behavior of FZ and HAZ seemed to be similar to that exposed by their respective base metals (REF). Nevertheless, the small difference in the mass magnetization values among base metals, fusion zones, and heat affected zones allowed the comparison of the NMI content present in these zones.

A clear comparison among HAZ and FZ graphs and their corresponding base metals (REF) could be done. Most of the RA content of both welded zones had been dissolved so that only non-metallic inclusions and ferromagnetic phases were present. The mass magnetization values of heat affected zones were slightly higher than those exhibited by their corresponding FZ, this behavior indicated that more paramagnetic phases (NMI) were present in the FZ of the steels. Similar magnetic behaviors are obtained by Amirthalingman et al. [4], they report an increment in paramagnetic phases as the magnetization values approach to zero. If the behavior of each of the TRIP steels was compared with either of both zones, then it was evident that steels with higher alloying elements tended to decrease their mass magnetization due to the presence of higher contents of NMI. Table 3 summarizes the REF, FZ, and HAZ magnetization saturation $M_s$ of the NMI non-containing samples of the spot welded TRIP steels.

**Table 3.** Saturation magnetization values of the non-containing austenite fusion zone and heat affected zone (FZ and HAZ $M_s(i)$) and non-containing (REF $M_s(f)$) samples.

| Steel | S1 | S2 | S3 | S4 | S5 | S6 |
|---|---|---|---|---|---|---|
| FZ $M_s(i)$ (Am$^2$/kg) | 217.1 | 207.44 | 193.4 | 184.49 | 178.9 | 173.31 |
| HAZ $Ms(i)$ (Am$^2$/kg) | 218.0 | 208.4 | 194.35 | 185.42 | 179.81 | 174.3 |
| REF $M_s(f)$ (Am$^2$/kg) | 218.3 | 208.8 | 195.6 | 186.1 | 180.4 | 175.1 |

The NMI contents in the FZ and HAZ were obtained from the magnetization curves using Equation (2). The NMI contents in the base metals were obtained by subtracting the µSXRD RA volume fractions to the RA + NMI volume fractions ($f_y$ Equation (1)) base metals values. Both results are shown in Figure 8. Previous research works [4,18,23–25] combine both techniques; X-ray and magnetic saturation to quantified the RA volume fractions and they all have found good correlation between the results of both techniques. The differences in the NMI content within the base metals between steels of similar chemical compositions was negligible, nevertheless, in steels with richer chemical compositions the percentage of the NMI content was significantly increased.

The formation of non-metallic inclusions in TRIP steels parent metals is inherent to their chemical composition and fabrication parameters. During melting, the total NMI content can be properly controlled. However, during spot welding, the formation of NMI in the fusion zone of TRIP steels cannot be accurately controlled yet. The NMI increase in the fusion zones of the steels is explained by the presence of alloying elements such as Al, Mn, and Si, these elements have a strong chemical affinity for oxygen. Hence, as the content of these elements was increased from steels S1 to S6, and in addition to the lack of a protecting atmosphere during the RSW process, the formation of non-metallic inclusions in the fusion zones of the steels was greater than in their parent metals. In addition to oxide type NMIs, other kind of inclusions such as sulfides, nitrides, and borides can be formed in the base metals and fusion zones due the richer chemical compositions of TRIP steels. However, oxide inclusion formation during RSW TRIP steels is greater than other kinds because of the high amounts of deoxidizers such as

silicon and aluminum present in the steels as was reported by Amirthalingam et al. [5], who states that both elements are added to suppress the formation of cementite, however their affinity for O entails to form oxides during welding, leaving the weld pool depleted of these elements. Then RSW elements present in the liquid pool combine with atmospheric oxygen and form oxides. Likewise, Grajcar et al. [6], finds that the presence of elements with strong chemical affinity with O (Al, Mn, Si) and the lack of a protecting atmosphere during a welding process result in forming a large number of NMI in the FZ. The oxides percentages and their types in the fusion zone depend on the chemical compositions of the steel. Steels S1, S4, and S6 which had higher aluminum contents were supposed to present higher contents of aluminum oxides than the rest of the steels. Similarly, steels with higher silicon contents were expected to present higher amounts of silicates in their fusion zones. On the other hand, NMI contents within the HAZ were reduced due to the thermal gradients that this zone experienced during the welding process. This result is in accordance with that reported by Grajcar et al. [6], who finds that the amount of NMI in the HAZ and BM of TRIP steels is significantly smaller than in the FZ.

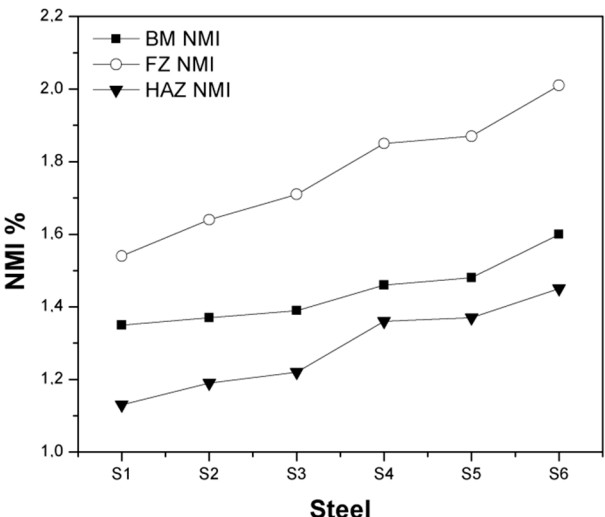

**Figure 8.** Non-metallic inclusion volume fraction in the TRIP steels base metals and welded zones.

### 3.5. Non-Metallic Inclusions Characterization in the Base Metals and Spot Weld Zones

Complex inclusions with different sizes were found in the base metals TRIP steels, their compositions and morphologies are shown in Figure 9. S1 steel (Figure 9a) showed an irregular shaped complex inclusion with average size of 4.5 μm. It was composed of elevated amounts of S, Mn, and Al, which indicated that manganese sulfide was present within the steel. MnS inclusions are found in steels deoxidized with an excess of Al [26]. Clusters of several small aluminum oxides with sulfur amounts and different morphology was found in S2 steel (Figure 9b). Figure 9c shows a MnS elongated inclusion with Si and Al rich zones, with an average area of 3 $\mu m^2$. Non-metallic inclusions present in steels S4 and S6 are shown in Figure 9d,f, respectively. Typical globular-like particles with diameters of 0.4 and 1 μm for steels S4 and S6 in the same order were present in both steels. Line scan spectrums indicated a notable increase in Fe and S contents on the particle of the S4 steel, with Si, P, and Mn contents. High Fe and O increments in addition to low amounts of C, Al, Si, Ti, Cr, and Mn were also registered for S6 steel, this made evident the presence of a complex FeO in the steel. Al–O inclusion with polyhedral morphology and average area of 2.5 $\mu m^2$ was found in S5 steel (Figure 9e), with high amounts of Fe and Si and low Ca and Mn. In general, all steels showed non-metallic inclusions with compositions and amounts of other elements inherent to the chemical compositions of the TRIP base metals present in solid solution and the elements used in the deoxidizing process. These inclusions were formed by precipitation as a result of homogenous reactions in the steels. They were composed

principally of oxides and sulfides and the reactions that formed them may have been induced either by additions to the steel or simply by changes in solubility during the cooling of the steel [26].

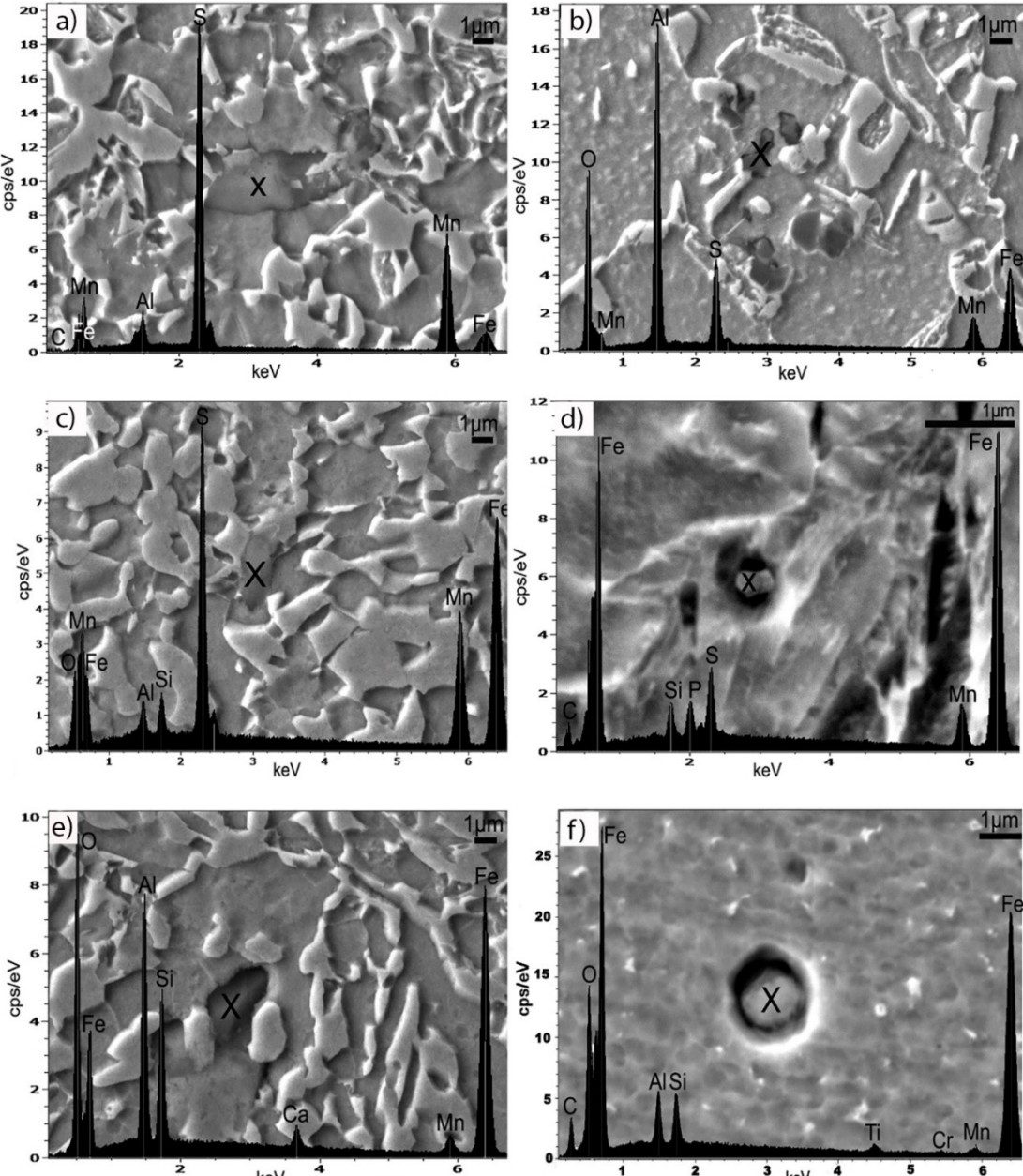

**Figure 9.** Energy-dispersive X-ray (EDX) microanalysis of non-metallic inclusions in the base metal TRIP steels: (**a**) S1 steel, (**b**) S2 steel, (**c**) S3 steel, (**d**) S4 steel, (**e**) S5 steel and (**f**) S6 steel.

Energy-dispersive X-ray (EDX) quantitative analysis of non-metallic inclusions present within the fusion zones of the TRIP steels are shown in Figure 10. Spectrum lines in all particles of all fusion zones indicated high presence of O. Fe intensities were bigger in fusion zones of steels with lower alloying contents as was the case of S1, S2, and S3 steels.

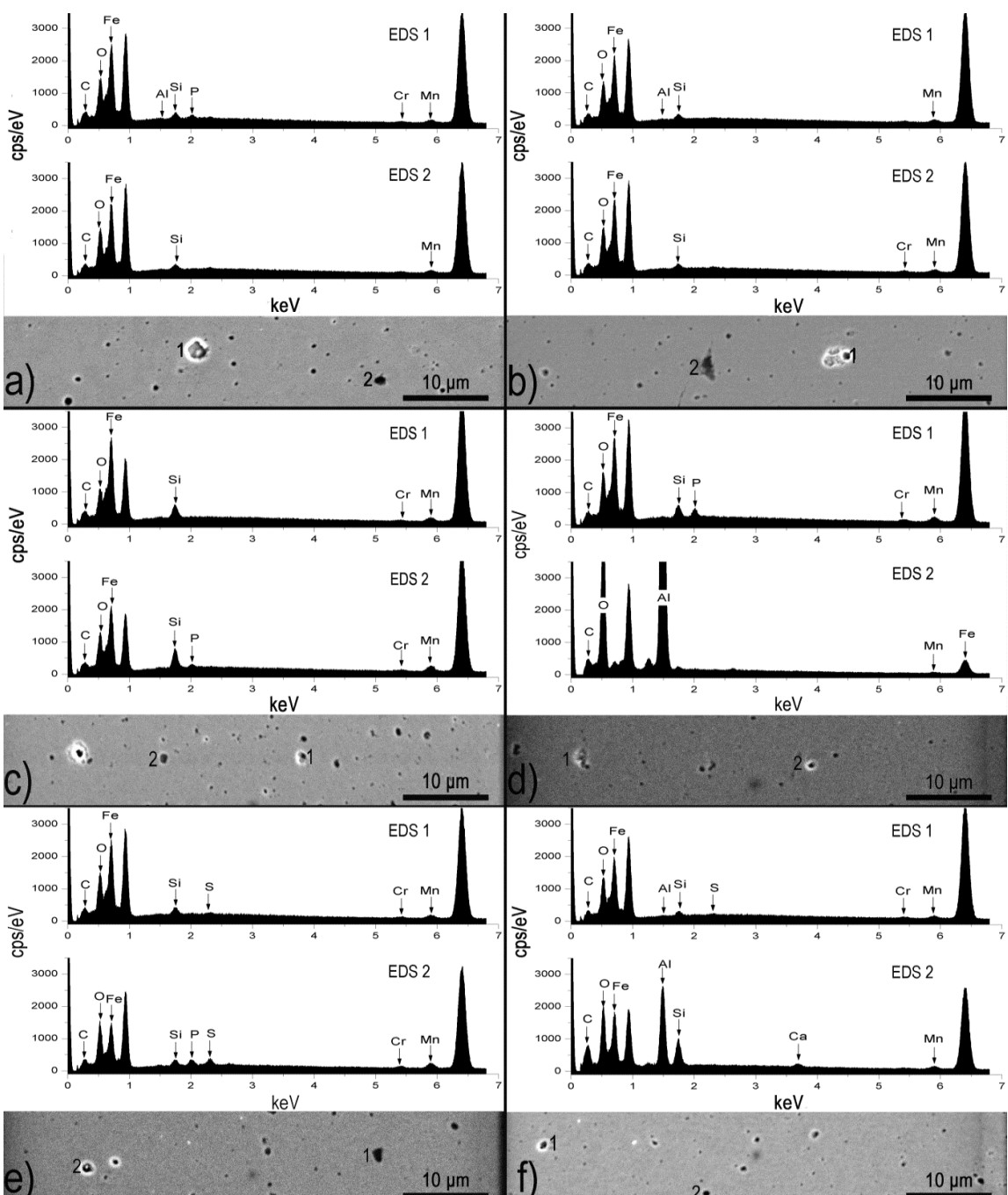

**Figure 10.** EDX microanalysis of non-metallic inclusions in the TRIP steels fusion zones: (**a**) S1 steel, (**b**) S2 steel, (**c**) S3 steel, (**d**) S4 steel, (**e**) S5 steel, and (**f**) S6 steel.

On the other hand, S4, S5, and S6 steels showed increased intensities in elements such as O, Al, and Si. It is clear from these spectrums that different compositional complex oxides were present in the fusion zones of the steels. Iron oxides with different amounts of Al, Si, P, Cr, and Mn were found in S1, S2, and S3 steels. Aluminum and silicon complex oxides with different amounts of S, Cr, Ca, and Mn were present in the fusion zones of S4, S5, and S6 steels. The higher amount of O within the fusion zones was easily explained by the fact that during the welding process there was not any protective atmosphere so that O present in the air was dissolved into the melting pool and it could combine with elements present in the liquid phase.

### 3.6. Mechanical Characterization of Spot Welds

The mechanical stability of weld joints was evaluated by means of the lap-shear tensile test. The average maximum loads and failure modes in combination with the average nugget diameters are shown in Figure 11. S6 steel showed the highest peak load with 22.95 ± 1.14 kN and an average diameter of 6.2 mm. S4 and S5 steels showed values of 21.86 ± 1.09 kN and 21.69 ± 1.08 kN, and diameters of 6.0 and 5.6 mm, respectively. S3, S1, and S2 steels presented values of 19.74 ± 0.98, 19.35 ± 0.96, and 18.25 ± 0.91 kN in the same order, and diameters of 5.7 mm for S3, 5.5 mm for S1, and 5.4 mm for S2.

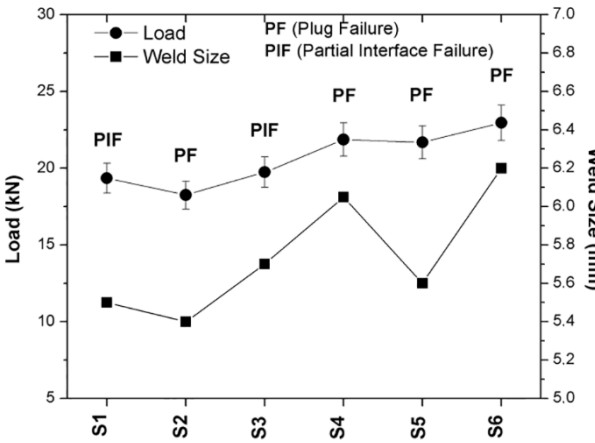

**Figure 11.** Peak load-weld size average relation of the spot welds of the TRIP steels.

A clear correlation can be made between the peak load and the diameter of the spot welds. Except for S5 steel, the rest of the steels showed an increase in resistance according with their weld nugget diameters as shown by Brauser et al. [21]. Russo Spena et al. [27], also report that mechanical resistance in TRIP/Q and P (Quenching and Partitioning) joints rise as the weld nugget size increases.

Plug mode failures were observed in S2, S4, S5, and S6 steels while S1 and S3 presented partial interfacial plug failure. Zuniga and Sheppard [28], finds that the failure mechanism for lap-shear sample on high-strength steels is localized necking in the base metal and near the boundary between HAZ and base metal. Similar results are reported by Khan et al. [29], who establish that the plug failure mode occurs at the external boundary of the HAZ. From the results revealed in Figure 1 and the failure mechanisms established in References [28,29], it is important to highlight that non-metallic inclusions developed within the fusion zones of the steels have minor effect when spot welds are mechanically tested under the lap-shear tensile test at the initial stage of deformation. However, non-metallic inclusions and RA within the HAZ have to be taken into account due to the fact that mechanical properties were enhanced by the first and reduced by the latter, and both phases were present in different percentage in the HAZ of the TRIP steels. In addition, the first stage of deformation will be supported by the HAZ of the weld nuggets if there are not preferential paths to propagate a failure due to segregation [29]. The non-metallic inclusions content decreased in the heat affected zones of all steels just as RA content did too. During RSW, the weld joint experienced different temperature ranges. Peak temperature above Ac1 was achieved in the weld nugget and decreased along from the middle of the nugget until the base material where temperature range was not enough to promote microstructural changes.

In the FZ, O present in the atmosphere combined with dissolved elements present in the weld pool leading to increase the NMI content, while retained austenite was nearly fully transformed to martensite due to the high cooling rates. Temperature range affected all welded zones, but HAZ the most. In this reduced weld zone the high differences in temperature were found to be above Ac3 and below Ac1, and could be divided in four sub-regions; (i) subcritical (SC) with temperature range below Ac1, (ii) inter-critical (IC) above Ac1, (iii) fine grained (FG) above Ac3, and iv) coarse

grained (CG) above Ac3 [29]. Non-metallic inclusions formation temperature ranges fell into the HAZ temperature ranges, so that some of the non-metallic inclusions present in the base metals tended to be dissolved during the RSW process leading to a decrease in their content in comparison with those present within the base metals. In the present research work, it was assumed that the NMI present in the HAZ of the TRIP steels were the original parent metal inclusions which were not dissolved during the welding process. In addition, most of RA present in the steels would also have dissolved in the FZ and HAZ sub-regions. However, in the SCHAZ the peak temperature was below Ac1 which was not high enough to transform the RA within this area and a lot less if the RA was stable. In addition, this area experienced lower cooling rates. In such way that if the heat treatment parameters were optimal for each steel, then RA fractions could be found within the HAZ due to this region change from the martensite generated by the high cooling rates to the base metal TRIP microstructure. As was the case of S5 and S6 steels whose transformation temperatures were closer to the optimal heat treatment temperatures, and RA traces were found in the HAZ of both steels. These steels also presented the higher contents of NMI in their BM, FZ, and HAZ samples. However, the small amounts of RA within their heat affected zones were stable enough to withstand higher mechanical load than the rest of the TRIP steels even when S5 steel had a small weld nugget diameter.

## 4. Conclusions

The present research work examined the microstructural development when resistance spot welding TRIP steels by correlating their post weld mechanical properties and base metal chemical compositions to the decomposition of RA and evolution of NMI in their weld zones. The main conclusions are listed as follows:

(1) Maximum stable retained austenite in TRIP steels was achieved by increasing alloying elements such as Si, P, and Mn, in addition to a properly designed TRIP heat treatment based on the chemical composition (transformation temperatures) of each TRIP steels.

(2) Non-metallic inclusions content within the fusion zones of the steels increased with increasing deoxidizing elements such as Si, Mn, and Al. On the other hand, some parent metal non-metallic inclusions were dissolved during the welding process in the heat affected zones of the steels, so that its content was decreased in comparison with their corresponding base metals.

(3) Traces of retained austenite could be found in the heat affected zone where the temperature of the resistance spot welding process was below Ac1 and base metal retained austenite was stable enough to withstand transformation due to temperature.

(4) The development of non-metallic inclusions and the degradation of retained austenite, both had detrimental effects over the post weld mechanical properties of TRIP steel. However, the decomposition of the later had a major effect due to that its absence allowed the load necessary to initiate fracture to be lower.

**Author Contributions:** Conceptualization, V.H.V.C., V.H.B.H., and C.M.Z.; investigation, V.H.V.C.; methodology, V.H.V.C.; supervision, G.A.G., I.M.G., V.H.B.H., and C.M.Z.; writing—original draft, V.H.V.C.; writing—review and editing, V.H.V.C., G.A.G., I.M.G., V.H.B.H., and C.M.Z.

**Funding:** This research was funded by Consejo Nacional de Ciencia y Tecnología (CONACYT), N.B. 254928.

**Acknowledgments:** Authors would like to thank the Coordinación de la Investigación Científica (CIC) of the Universidad Michoacana de San Nicolás de Hidalgo (UMSNH-México) for the support during this project (CIC-UMSNH-1.8). Víctor H. Vargas Cortés studies were sponsored by the National Council on Science and Technology (Consejo Nacional de Ciencia y Tecnología-México) and would like to thank them for the support during this project N.B. 254928.

**Conflicts of Interest:** The authors declare no conflict of interest.

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
