# Peer review of "Effect of Retained Austenite and Non-Metallic Inclusions on the Mechanical Properties of Resistance Spot Welding Nuggets of Low-Alloy TRIP Steels"

_metals, doi:10.3390/met9101064_

Round 1

Reviewer 1 Report

This is an interesting report concerning the weldability of TRIP steels. Through the use of a variety of techniques, the authors' work reveals the effect of chemical composition on the outcome of spot-welding, by assessing the evolution of retained austenite and non-metallic inclusions and their impact on mechanical properties. I believe that the manuscript can be accepted to the journal publication. I would only make the following minor remarks. 

1// the authors stated that the volume fraction of retained austenite varies with the silicon content. The Si content could be reported in Fig. 6 to complement the information reported there.

2// What is the value of beta in Eq. 1?

3// There are distinct mechanisms of failure in mechanical tests, it would be useful to have some images of fractured surfaces 

Reviewer 2 Report

The article deals with a rather important problem of quality of spot welds of high strength steels. Joints of AHSS are still going topic of research, as these steels have a very special microstructures that are easily destroyed by heat input during joining.

There are several points that need to be clarified:

Please provide an explanation of the idea behind alloying concepts of the six steel used. Highlight the difference between them, as in recent tab.1 it is very difficult to distinguish what is random scatter and what is intended difference. For example were Cr, Ni, Mo contents intended to differ among the samples or was it just deviation of the same prescribed value? Could you also name the samples rather by important alloying elements rather than 1-6? Had sample 1 really 0.0025 N while all other samples had 0.01 or 0.02%? If so, explain why.

The temperature of intercritical annealing was 812°C (line 79, Fig. 1). However, this is not exactly true. According to tab. 1, this temperature would really be in intercritical area only for S3, S5 and S6 steel. It corresponds to AC1 temperature of S4 steel and it is in fully austenitic region for S1 and S2. Due to this fact, different initial microstructures has to be created in various steel by the first annealing step. Is it really fair to compare the steel between each other than? Wouldn’t it be more suitable to use several soaking temperatures which would ensure that the same ferrite / austenite ration is achieved in all used steels after the first annealing step? Could you correct your statement and comment on above mentioned facts? 

Could you add the information about cooling rate used for cooling from “intercritical” annealing temperature to bainitic hold? Since the treatment was monitored by a thermocouple (line 83) this value should be available. Was used slat bath heated (and at what temperature)?

Lines 140-141 state that: “ …consist of intercritical annealing to transform the initial microstructure into austenite”. While the following sentence states (correctly) that intercritical annealing is in ferrite + austenite zone. Please rephrase tis part. The point of intercritical annealing is to create austenite AND FERRITE in various ratios, as you are in two-phase region.

Fig. 3 proves that little or non intercritical annealing was in fact carried out in most of the steel. Intercritical (polygonal) ferrite can be clearly seen only in 3a). All other microstructures definitely do not have ferrite matrix as mentioned in line 131. They are rather bainitic microstructures (with exception of 3a) with larger areas of bainitc ferrite and perhaps occasional fine ferrite grains in Fig. 3b, c. Proper caption of Fig. 3 is missing, describing what steels are in a) –f).

The same problem with missing captions describing images a)-f) is in Fig. 4, 5 and 7 (a-d).

Line 142 is defining the suitable temperature as “20 to 30°above AC1”. Would this statement be really generally valid, as the phase fraction “20 to 30°above AC1” could differ in different steels based on the width of intercritical temperature interval? Either add a particular steel for which this statement is true, or define rather the most suitable ferrite and austenite fraction.

Line 148: should be “…rapid cooling causes the…” rather than “…rapidly cooling causes to the…”

If the trend in metallic inclusion number is quite clear already from macro-images of the welds (Fig. 4, line 168), what is the point of all the subsequent rather elaborated analysis? Could you quantify the inclusion fraction by image analysis of higher resolution optical micrographs? Would this method be sufficient to compare the inclusion fraction in a set of samples?

It is stated in line 169 that random distribution of inclusions was observed. However it looks from Fig. 4 b) and d) that inclusions were predominantly placed at FZ/HAZ boundary. Could you comment on this? What is the darker line separating FZ and HAZ in Fig. 4a) and f), could you explain it and why is it visible only at these two images?

Fig. 6 – the last sample on horizontal axe should be S6 rather than the second S5?

Could you really work with retained austenite volume fraction determined by micro X-ray diffraction analysis as being equal to retained austenite volume fraction which would be determined by magnetization method? The methods evaluated very different volumes of metal and no mention or documentation of homogeneity of retained austenite distribution across the thickness of used sheets and across the depth of the weld metal were provided. Could you comment on this?

Line 303 – what is dendritic about oxides in Fig. 9b? Only the group of several small oxides is visible?

Fig. 10 Could you provide element continents to individual inclusions rather than the spectra? It is rather difficult to evaluate the differences among the many spectra.

What is the “plug failure” mentioned in chapter 3.6.? Could you provide images of both typed of fracture mentioned there and point out the difference?

Why is weld size for sample S5 not following the trend of other samples?  (Fig. 11)

If it is already known that the resistance of joint increase with nugget size (line 350), how could you compare performance of samples with varying nugget sizes? Please comment on this.

Reviewer 3 Report

Its an interesting research about the RSW of selfmade TRIP Steels.

In my opinion title should contain resistance, there are a lot of spot welding forms.

Some general comments:

In the title stands "Welding Nuggets " in the manuscript FZ, recommend to us WN in the manuscript also.

For low alloy steels the HAZ of a RSW-ed nugget consists of three zones UCHAZ, ICHAZ and SCHAZ. To evaluate the investigations about the "HAZ", the spot size/exact are of investigation is needed. e.g. spot size and location of XRD which HAZ zone was measured,or an average was measured of all three?

Microstructure images for the joints zones would be a great support for the XRD measurements.

To evaluate the mechanical properties of the joints, the base materials strength, and mechanical properties are needed, put them into the manuscript!

Also at least a hardness line profile on the joint would be needed.

Specific remarks, questions, comments are also listed in the manuscript_with_reviewers_comments

I think with proper corrections the manuscript could fulfill the publication criteria for Metals.
